# Establishment of soil strength in a nourished wetland using thin layer placement of dredged sediment

Brian D. Harris[1,2]ᴼ *, Donnie J. Day[1,3]ᴼ, Jack A. Cadigan[1]ᴼ, Navid H. Jafari[1]‡, Susan E. Bailey[1]‡, Zachary J. Tyler[2]‡

**1** Department of Civil and Environmental Engineering, Louisiana State University, Baton Rouge, Louisiana, United States of America, **2** US Army Corps of Engineers, Research and Development Center, Vicksburg, Mississippi, United States of America, **3** Department of Oceanography and Coastal Sciences, Louisiana State University, Baton Rouge, Louisiana, United States of America

ᴼ These authors contributed equally to this work.
‡ These authors also contributed equally to this work.
* bharr96@lsu.edu, brian.d.harris@erdc.dren.mil

**Data Availability Statement:** All data has been supplied in the accompanying excel sheet. Sheet 1 contains all cone penetrometer data and is set up with a consistent depth interval in column A and

## Abstract

Coastal wetlands are experiencing accelerated rates of fragmentation and degradation due to sea-level rise, sediment deficits, subsidence, and salt-water intrusion. This reduces their ability to provide ecosystem benefits, such as wave attenuation, habitat for migratory birds, and a sink for carbon and nitrogen cycles. A deteriorated back barrier wetland in New Jersey, USA was nourished through thin layer placement (TLP) of dredged sediment in 2016. A field investigation was conducted in 2019 using a cone penetrometer (CPT) to quantify the establishment of soil strength post sediment nourishment compared to adjacent reference sites in conjunction with traditional wetland performance measures. Results show that the nourished area exhibited weaker strengths than the reference sites, suggesting the root system of the vegetation is still establishing. The belowground biomass measurements correlated to the CPT strength measurements, demonstrating that shear strength measured from the cone penetrometer could serve as a surrogate to monitor wetland vegetation trajectories. In addition, heavily trafficked areas underwent compaction from heavy equipment loads, inhibiting the development of vegetation and highlighting how sensitive wetlands are to anthropogenic disturbances. As the need for more expansive wetland restoration projects grow, the CPT can provide rapid high-resolution measurements across large areas supplying government and management agencies with vital establishment trajectories.

## Introduction

Coastal wetlands provide a variety of vital ecological services, including fish and wildlife habitat, water filtration, carbon and nutrient sequestration, and flood and storm protections [1–7]. However, due to the sea-level rise, hydrologic and sediment restrictions, rates of wetland

test data in the remaining columns. Sheet 2 contains the grain size data. Column 3 contains the elevation data. Column 4 contains belowground biomass, bulk density, and water content values for all cores.

**Funding:** The author(s) received no specific funding for this work.

**Competing interests:** The authors have declared that no competing interests exist.

fragmentation and loss have accelerated. Thin layer placement (TLP) is a common restoration management strategy used throughout the Gulf, Atlantic, and Pacific coasts of the United States that focuses on improving biotic and abiotic environmental conditions through pumping hydraulically dredged sediments onto the marsh platform [8–17]. While TLP appears to be an effective restoration strategy, there is limited information on the effectiveness of dredged sediments on vegetation establishment and wetland soil strength.

The application of dredged sediments to deteriorating wetlands increase marsh elevation and improve soil aeration in the root zone, thereby increasing redox potentials (Eh), plant productivity and soil accretion allowing marshes to keep pace with relative sea-level rise [9, 16, 18, 19]. However, quantifying belowground soil stability or strength following TLP is still poorly understood and despite being a crucial parameter in the prediction of wetland sustainability (i.e., erosion, ponding, collapse, uprooting) [20, 21]. Only recently has methodology for quantifying wetland strength using a cone penetrometer been standardized [21]. As a result, the following study implements a cone penetrometer test (CPT) to measure belowground soil strength against accepted trajectory performance measures to determine the influence of TLP on biotic and abiotic soil properties.

The CPT is a common method utilized by geotechnical engineers to define stratigraphy and engineering behavior of soil for infrastructure projects (e.g., levees, dams, and bridges) [22]. The standard CPT measures tip resistance, sleeve friction, and pore-water pressure [23], but additional modules can be added to measure soil moisture, resistivity, and temperature [21]. Along with the ability to provide a wider range of data, a major advantage of the CPT over other field methods (e.g., handheld shear vane or torvane) is they provide a continuous resistance profile with depth allowing for a better estimation of subsurface site stratigraphy. In addition, CPTs can be conducted faster than field vane tests which can allow for a more robust spatial data set. The implementation of CPTs in wetlands has not been common due to instrumentation limitations but has become more common over the past decades. In particular, they have been utilized to identify groundwater recharge zones within Massachusetts, USA [24] and to better understand differences in salt marsh stability in coastal Louisiana, USA [20]. Most recently [21], developed a CPT to understand the vertical and spatial variations of geotechnical properties of salt marshes in Louisiana, USA.

In March 2016, the United States of America Corps of Engineers (USACE) Philadelphia District (NAP) began restoration of coastal wetlands in New Jersey, USA via TLP. In total, the restoration deposited 34,405 m3 (45,000 yd3) of dredged sediments from the New Jersey Intracoastal Waterway (NJICW) [19] across five containment sites. The sediment was contained using coir logs that were slashed after stabilization of the dredged sediment (approximately 6 months) to expedite coir log deterioration. Placement target elevations, based on tidal and biological references, ranged from 0.73 to 0.91 m (2.4 to 3.0 ft) NAVD88 [25, 26]. Sediment thicknesses ranged between 5.2 cm to 9.5 cm in the vegetated areas and 32.5 cm to 82.5 cm in the open water features six months after dredging was completed [16].

In this 2019 study, two 100 m transects were conducted within one of the 2016 TLP containment sites moving downstream of the dredge outfall. The first transect traversed a previously ponded section and the second transect traversed a tidal creek. The results are compared to a reference site 500 m northeast of the nourishment site. This study is the first to evaluate the benefits of dredged sediment on a wetland using a CPT and traditional trajectory performance measurements to quantify soil strength gain within a wetland nourishment. These findings can serve as a benchmark for future restoration projects by demonstrating useful establishment monitoring practices for coastal stakeholders.

## Background

### Site description

The study site is located near Avalon, New Jersey, USA (Fig 1). The sites are within a 17 km2 tidal marsh complex adjacent to Great Sound. Freshwater input to the site is limited, with the nearest rivers located approximately 26 km (Great Egg Harbor River) and 56 km (Mullica River) to the north [27]. The surficial sediments are primarily comprised of Holocene era salt marsh and estuarine deposits (organic silts and clays with sand) to a maximum depth of 18 m [28, 29], which pinches out from east to west [27]. The Holocene sediments are underlain by a Pleistocene sand deposit, which is approximately 37 m thick with some interbedded silts [27, 29].

Wetland degradation of the area was identified in the rapid transition of vegetated marsh platforms to un-vegetated shallow pannes through erosion and vegetation stress over the course of decades [16, 19, 30]. The primary vegetation located across the wetland was a low-form of *S. alterniflora* that transitions to tall-form *S. alterniflora* as elevation decreases near ponds and tidal creeks (Fig 1). The vegetated areas are located at an average elevation of 0.61 m NAVD88 (±0.20 m) while shallow open water pannes were found at an average elevation of

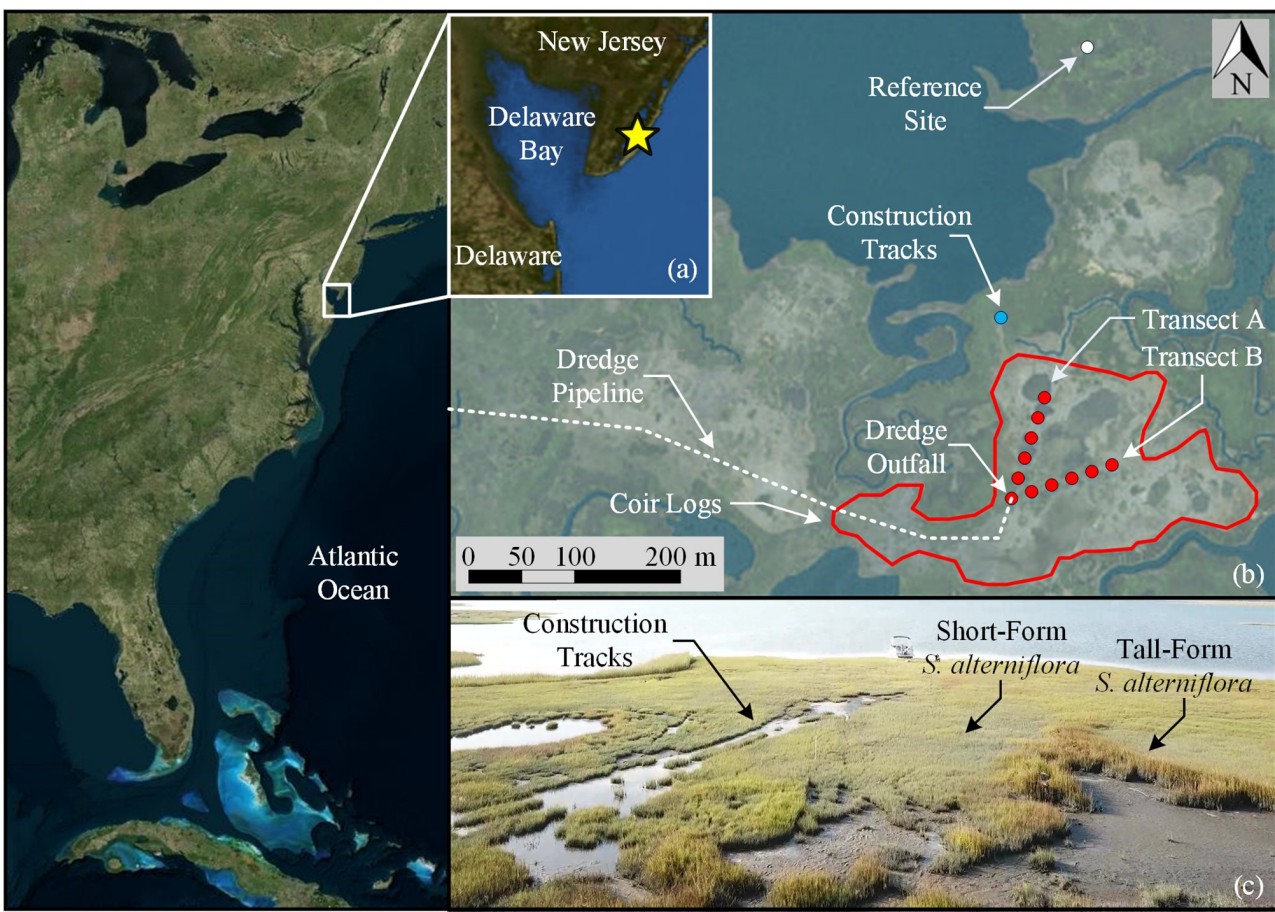

**Fig 1. Overview of sediment nourishment study site in Avalon, New Jersey, USA.** (a) Location of the study site. (b) Investigation layout, coir log containment, dredged outfall, and pipeline locations. Aerial images from The National Map Orthoimagery courtesy of the U.S. Geological Survey (2020). (c) An image showing tall- and short-forms of *S. alterniflora*. Image taken by author.

0.26 m NAVD88 (±0.13 m) [16]. The site experiences a semidiurnal tidal range of 1.39 m with a MHHW of 0.74 m NAVD88 [30].

## Methodology

This field study was comprised of two transects within one of the nourishment sites: (1) a transect that traversed a ponded section, herein referred to as "Transect A", and (2) a transect that traversed a tidal creek, herein referred to as "Transect B" (Fig 1). Both transects extended 100 m from the dredge outfall with 20 m sample spacing and were selected to show contrasting starting points on the implications of establishment. The field access and sampling collection was approved and overseen by The Wetlands Institute. In addition, the individuals displayed in this manuscript have given written informed consent to publish these case details.

### Cone penetrometer

The CPT utilized was specifically developed for use in ultra-soft wetland soils, commonly found in coastal Louisiana, that incorporates both geotechnical parameters (tip resistance, sleeve friction, and pore pressure) and abiotic parameters (soil moisture, electrical resistivity, and temperature) [21]. These ultra-soft soils have high water content and compressibility, may be underconsolidated, and are undergoing self-weight consolidation [31]. For this study, sleeve resistance is considered analogous to shear strength since the goal of this methodology is to provide comparable values as opposed to an engineering design variable. This field equipment consists of a cone piezometer, potentiometer, and a backpack mounted Data Acquisition System (DAS) capable of being manually operated by a three-person crew (Fig 2a). A modified sleeve with 5.2 cm length, perpendicular fins (Fig 2b) was used throughout this study to increase the resolution of soil resistance throughout the vegetated root mass and into the softer organic clay. Four (4) tests were conducted within 1 m2 at each site to better capture substrata variations in root establishment.

The CPT soundings are conducted manually, with a target standard penetration rate of 2 cm/s [23]. However, the push speed can vary and occasionally stops if a stiff layer is encountered. A stop in penetration causes a drop to zero push speed (see Fig 3a) and subsequent increase in tip and sleeve resistance values (see Fig 3b at locations with stops) when cone

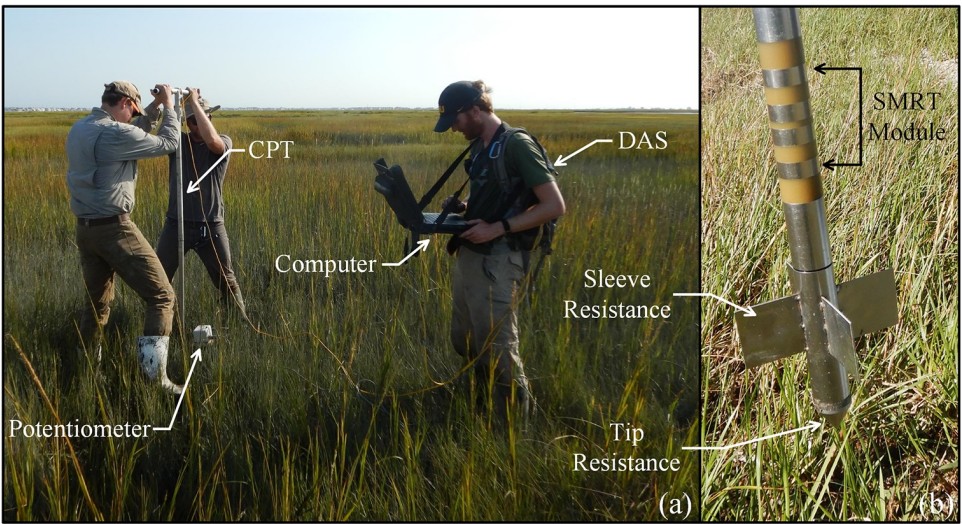

**Fig 2. (a) Performing a cone penetrometer test and (b) cone penetrometer components.**

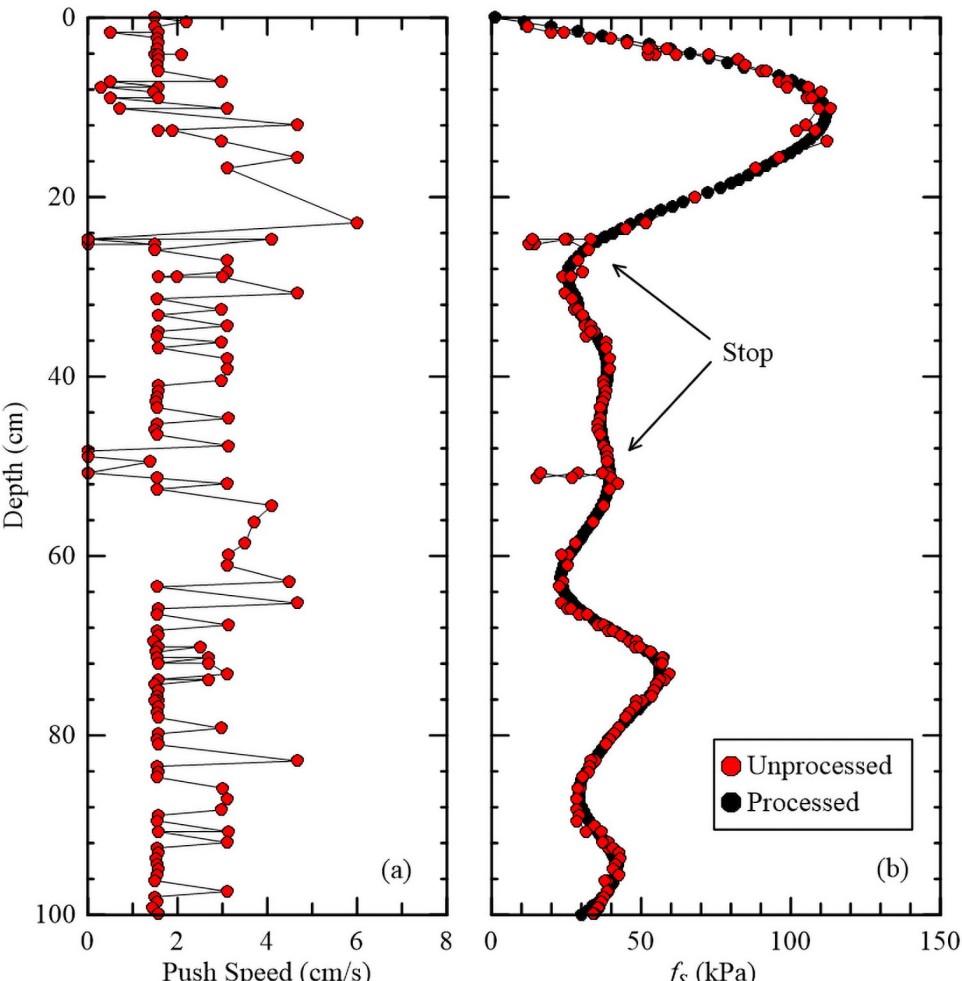

**Fig 3. Example of (a) push speed in relation to the (b) sleeve resistance ($f_s$) values.**

penetration recommences due to soil consolidating around the cone. A Matlab script removes these stops through the application of a Savitzky-Golay filter [32], which applies convolution to smooth the data without distortion of the signal tendency [33]. The push speed in relation to unprocessed and processed sleeve resistance data is shown in Fig 3. The average push speed in Fig 3a is 1.5 cm/s. The unprocessed data (red circles in Fig 3b) show distinct stops at 25 cm and 50 cm. The processed data (black circles in Fig 3b) remove these stops and interpolate between the less dense red circles from 10 cm to 20 cm. The paucity of data in this depth range is due to the significant force required to push through the root mat, causing the CPT to rapidly break through into the underlying soil layer at a rate faster than the sampling frequency of the data acquisition system.

## Field samples

Along the transects, soil samples were collected using polyvinyl chloride (PVC) cores and a Russian Peat Corer to validate the CPT measurements with traditional performance measures. Soil cores were collected by inserting the PVC pipe (diameter 15 cm) to an approximate depth of 30–35 cm. Samples from these cores were field extruded and cut into 5-cm

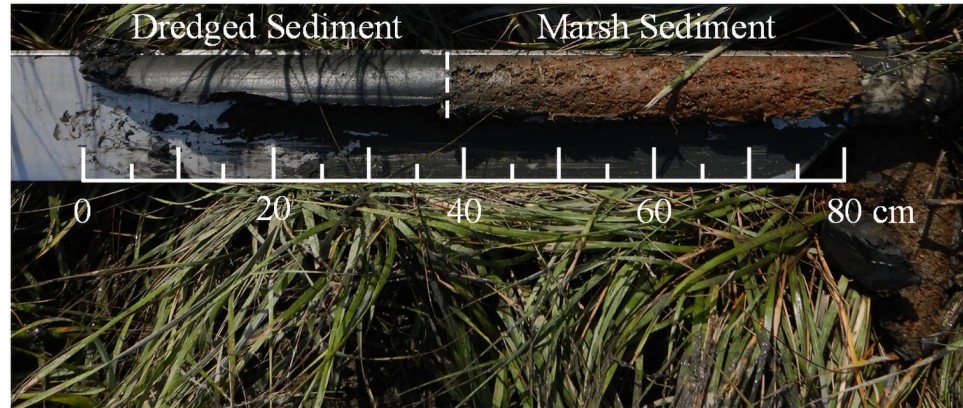

**Fig 4. Example of Russian Peat Core sample from Transect A at 60 m.** The shallower portion (left) showing the dredged sediment and the deeper (right) showing the vegetated marsh sediment.

sections, bagged, shipped on ice, and stored at 4°C until processing. The samples were processed to determine belowground biomass, bulk density, and moisture content with depth. Four (4) Russian Peat Corer samples were taken at each site within the TLP placement area to delineate between the dredged sediment (dark gray and homogenous) and vegetated marsh sediment (brown). An example of the transition from the brown organic to dark grey mineral layer from the Interior at 60 m is shown in Fig 4. The dredged sediment depth was recorded and collected in bags to determine the grain size of the deposited material in accordance with [34].

## Results

### Sediment redistribution

Post TLP nourishment, researchers from the USACE Engineering Research and Development Center (ERDC) collected dredged sediment samples along Transect A to determine the change in grain size moving down gradient of the dredge outfall. A comparison between the percent of fine-grained material (<0.075 mm) from the 2016 results and the 2019 investigation is shown in Fig 5. The 2016 sampling showed that as distance from the dredge outfall increased, the percentage of fine-grained sediment increased. This is due to coarser material (i.e., sands) falling out of suspension faster than the finer material (i.e., silts and clays). However, the same trend is not evident in 2019, which indicates a more constant percentage of fines across the transect. This redistribution of fines could be attributed to vegetation-driven accretion or movement of the sediment due to hydrodynamic processes and bioturbation [13].

### Transect A: Soil strength

Along Transect A, short-form *S. alterniflora* appeared to be the dominant vegetation type, extending out to 40 m from the dredge outfall then transitioned to a mudflat until 80 m where it became sparsely vegetated tall-form *S. alterniflora* until 100 m. The shear strength ($\tau$) ± 1 standard deviation (SD) with depth and dredged sediment thicknesses for each site through Transect A are shown in Fig 6, and Table 1 summarizes the geotechnical and ecological data. At the dredge outfall and 20 m locations, dredged sediment was recorded down to 5 cm and the sites had approximate shear strengths of 165 kPa. As the amount of placed dredged

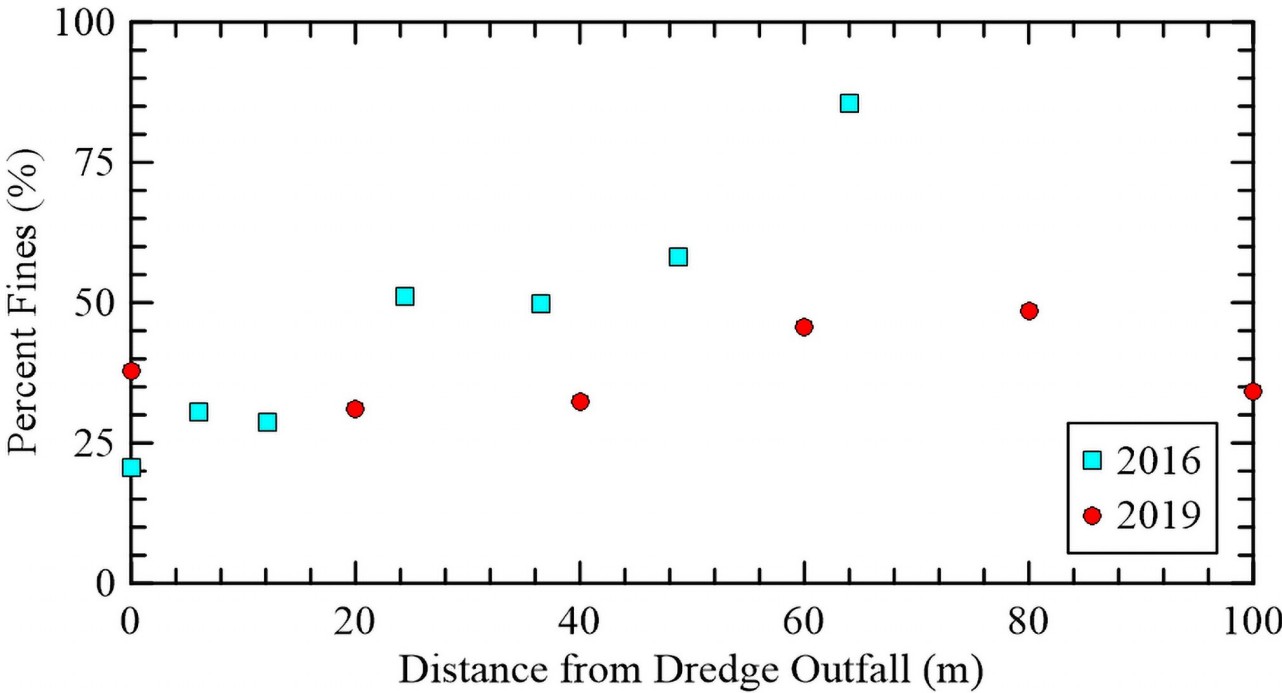

**Fig 5. Percent of fine grained material (<0.075 mm) with distance from the dredge outfall from field investigations.**

sediment increases, the corresponding shear strengths decreased with 10 cm of sediment and a strength of 126 kPa at 40 m and ultimately found strengths lower that 90 kPa when the dredged sediment depths were greater than 30 cm. Across Transect A, the underlying soil shear strength averaged 42 kPa with depth.

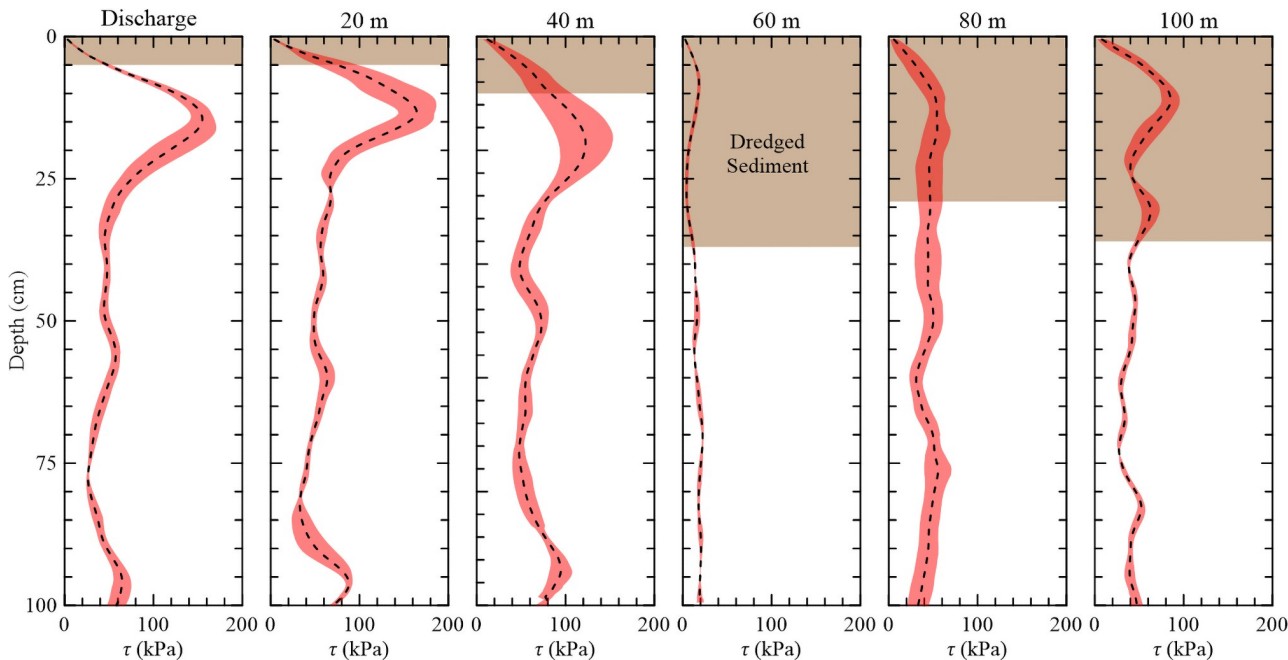

**Fig 6. Average shear strength and dredged sediment depths along Transect A moving away from the discharge.** Shaded regions represent ±1 SD.

**Table 1. Site description and belowground characteristics along Transect A.**

| Site | Surface Type | Dredged Sediment (cm) | Belowground Biomass | | | Underlying Soil |
| --- | --- | --- | --- | --- | --- | --- |
| | | | Peak $\tau$ (kPa) | Peak Depth (cm) | Root Depth (cm) | Avg. $\tau$ (kPa) |
| Discharge | Short | 5 | 159 | 15 | 29 | 43 |
| 20 m | Short | 5 | 172 | 14 | 25 | 52 |
| 40 m | Short | 10 | 126 | 21 | 30 | 55 |
| 60 m | Mudflat | 37 | 18 | 9 | - | 17 |
| 80 m | Tall | 29 | 55 | 13 | - | 44 |
| 100 m | Tall | 36 | 87 | 11 | 22 | 40 |

Surface type is Short-form *S. alterniflora* (Short), Tall-form *S. alterniflora* (Tall), and Mudflat.

## Transect B: Soil strength

Along Transect B, short-form *S. alterniflora* appeared to be the dominant vegetation type, extending out to 40 m from the dredge outfall then transitioned to tall-form SA at 60 m. Around 80 m, the transect traversed a tidal creek until 100 m where it became a sparsely vegetated tall-form *S. alterniflora*. The shear strength ($\tau$) ± 1 standard deviation (SD) with depth and dredged sediment thicknesses for each site through Transect B are shown in Fig 7, and Table 2 summarizes the geotechnical and ecological data. At the dredge outfall, dredged sediment was recorded down to 5 cm and the site had a strength of 159 kPa. As with Transect A, the peak shear strengths decreased as the dredged sediment thickness increased with 10 cm of sediment and a strength of 146 kPa at 40 m and then strengths less than 119 kPa for dredge sediment thicker than 31 cm. Across the transect, the underlying soil shear strength averaged 36 kPa with depth.

## Reference site

Within the Reference Site, CPTs were conducted within a short- and tall-form of *S. alterniflora*, with short-form being the dominant vegetation. The reference area was assumed as

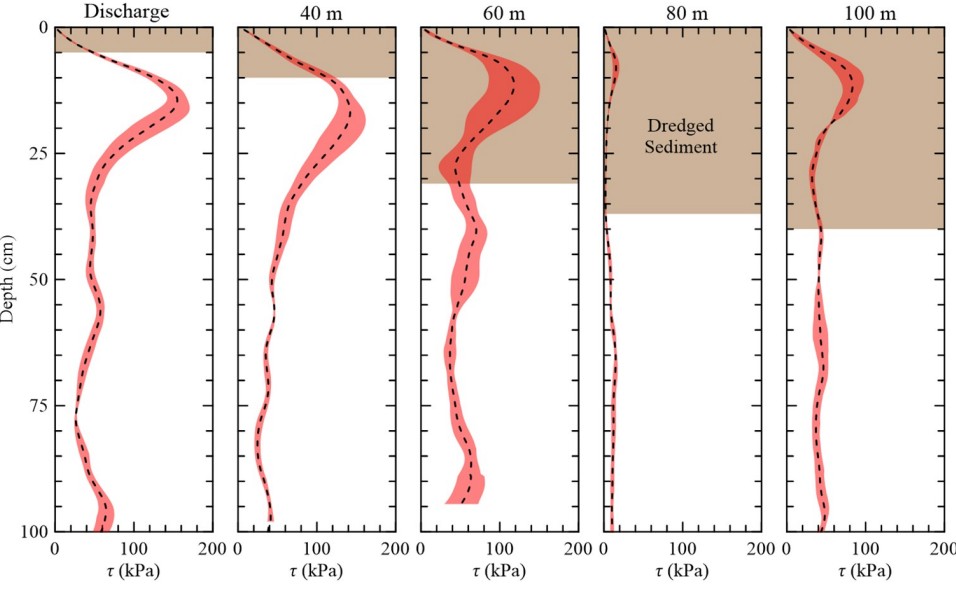

**Fig 7. Average shear strength and dredged sediment depths along Transect B moving away from the discharge.**
Shaded regions represent ±1 SD.

**Table 2. Site description and belowground characteristics along Transect B.**

| Site | Vegetation Type | Dredged Sediment (cm) | Belowground Biomass | | | Underlying Soil |
|---|---|---|---|---|---|---|
| | | | Peak $\tau$ (kPa) | Peak Depth (cm) | Root Depth (cm) | Avg. $\tau$ (kPa) |
| Discharge | Short | 5 | 159 | 15 | 29 | 43 |
| 20 m | Short | 5 | | | | |
| 40 m | Short | 10 | 146 | 17 | 36 | 36 |
| 60 m | Tall | 31 | 119 | 11 | 23 | 49 |
| 80 m | Mudflat | 37 | 15 | 9 | 18 | 11 |
| 100 m | Tall | 40 | 84 | 12 | 24 | 40 |

Surface type is Short-form *S. alterniflora* (Short), Tall-form *S. alterniflora* (Tall), and Mudflat.

stable (i.e., no discernable deterioration) over the past three decades based on time-lapse aerial images. Due to time constraints, all coring equipment was being utilized within the nourished area so no soil cores were collected. The average shear strength values ±1 SD bands for both forms of vegetation are shown in Fig 8. For the short-form, the peak shear strength of 263 kPa was found at a depth of 11 cm and the root depth was recorded down to 40 cm. The tall-form

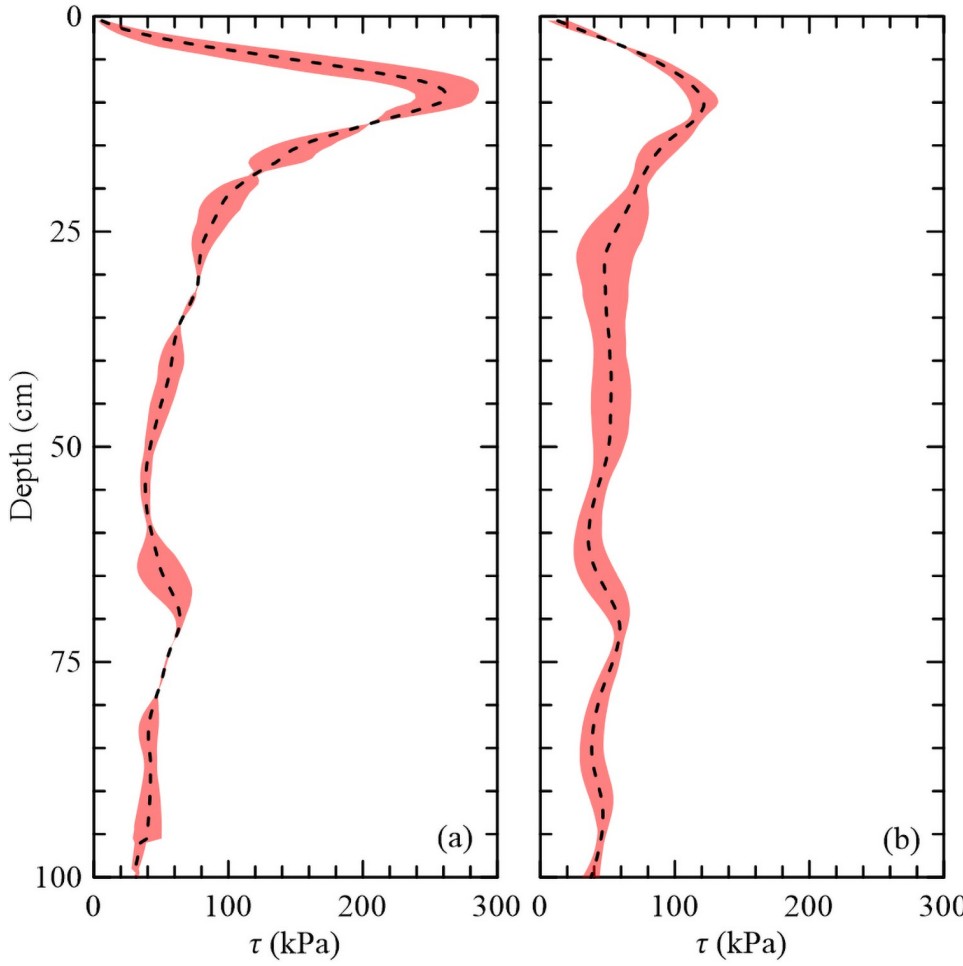

**Fig 8. Average shear strength at the reference sites.** (a) Short- and (b) tall-form *S. alterniflora* sites. Shaded regions represent ±1 SD.

of *S. alterniflora* was found at a lower elevation than the short-form, within a shallow confined pond. The peak shear strength of 123 kPa was found at a depth of 11 cm and the vegetation influence was recorded down to 30 cm. At the short- and tall-form sites, the underlying soil exhibited an average shear strength of 46 kPa. The tall-form root strength was weaker than the short-form likely due to the lower elevation resulting in a longer inundation period. The effect of roots on strength was only observed down to 30 cm for the tall-form when compared to the short-form of 40 cm root zone.

## Performance measures

To quantify the effectiveness of the CPT against other performance measures, the below-ground biomass, bulk density, and moisture content were measured at each site along the transects and compared to the shear strength profile. Shear strength and belowground biomass exhibited analogous trends with depth, increasing from the marsh surface to a depth of 13 cm where peak belowground biomass is located and then a similar decrease trend with depth to 30 cm, Fig 9.

The larger shear strengths (>75 kPa) correlated to samples of higher belowground biomass (>0.04 g/cm3), Fig 10. The weaker shear strengths were generally found within the dredged sediment which exhibited lower amounts belowground biomass. It is postulated that as the dredged sediment is further established, the presence of belowground biomass will increase the strength to resemble the measurements within the native "marsh sediment".

A summary of shear strength as it relates to belowground biomass, bulk density, and moisture content plots, divided into varying thicknesses of dredged sediment: ≤5 cm, 5–10 cm, 10–30 cm, and >30 cm are shown in Fig 11. The greatest shear strengths of 160 kPa were found in areas of ≤5 cm dredged sediment thickness with a peak depth of 13 cm, and the lowest shear strength values at peak depth were in areas >30 cm (Fig 11a). The shaded regions in the CPT profiles illustrate the variability, and it signifies that the <5 cm and 5–10 cm thicknesses exhibit less scatter (i.e., uncertainty). The greatest belowground biomass contribution of ~0.8 g/cm3 was found in areas that received ≤5 cm of dredged sediment (Fig 11b). The same trend was observed in the 5 to 10 cm thickness but with lower belowground biomass values. In contrast, the lowest belowground biomass contribution of 0.01 g/cm3 was found in areas of that received greater than 30 cm TLP because of their lower elevation and hence more deteriorated vegetation. Bulk densities were highest in areas that received >30 cm TLP, which is consistent to bulk densities of inorganic sediment [35]. The bulk densities for less than 10 cm TLP exhibit organic-rich soils [36]. At higher TLP thicknesses, bulk densities are approximately 0.6 g/cm3 near the surface for about the first 15 cm, but gradually decrease to bulk densities similar to that of the < 10 cm TLP (Fig 11c). This behavior could be a result of mixing of the dredged sediment with native sediment and provides another means to understand the trajectory of TLP. The moisture content [*w (%)*] behaves in direct contrast to bulk density because organic material inherently contains more pore water. The moisture content tends to increase with depth to more than 500% for the ≤5 cm and 5–10 cm sites, yet remained at constant with depth in the areas that received >30 cm of sediment (Fig 11d). These higher moisture contents are a key indicator of greater biomass establishment in areas receiving less than 10 cm of sediment whereas greater sediment inputs have lower moisture contents due to lower concentrations of organic material.

## Discussion

The purpose of this study was to investigate the usefulness of applying a CPT to investigate the establishment of wetland shear which can provide decision makers with validation

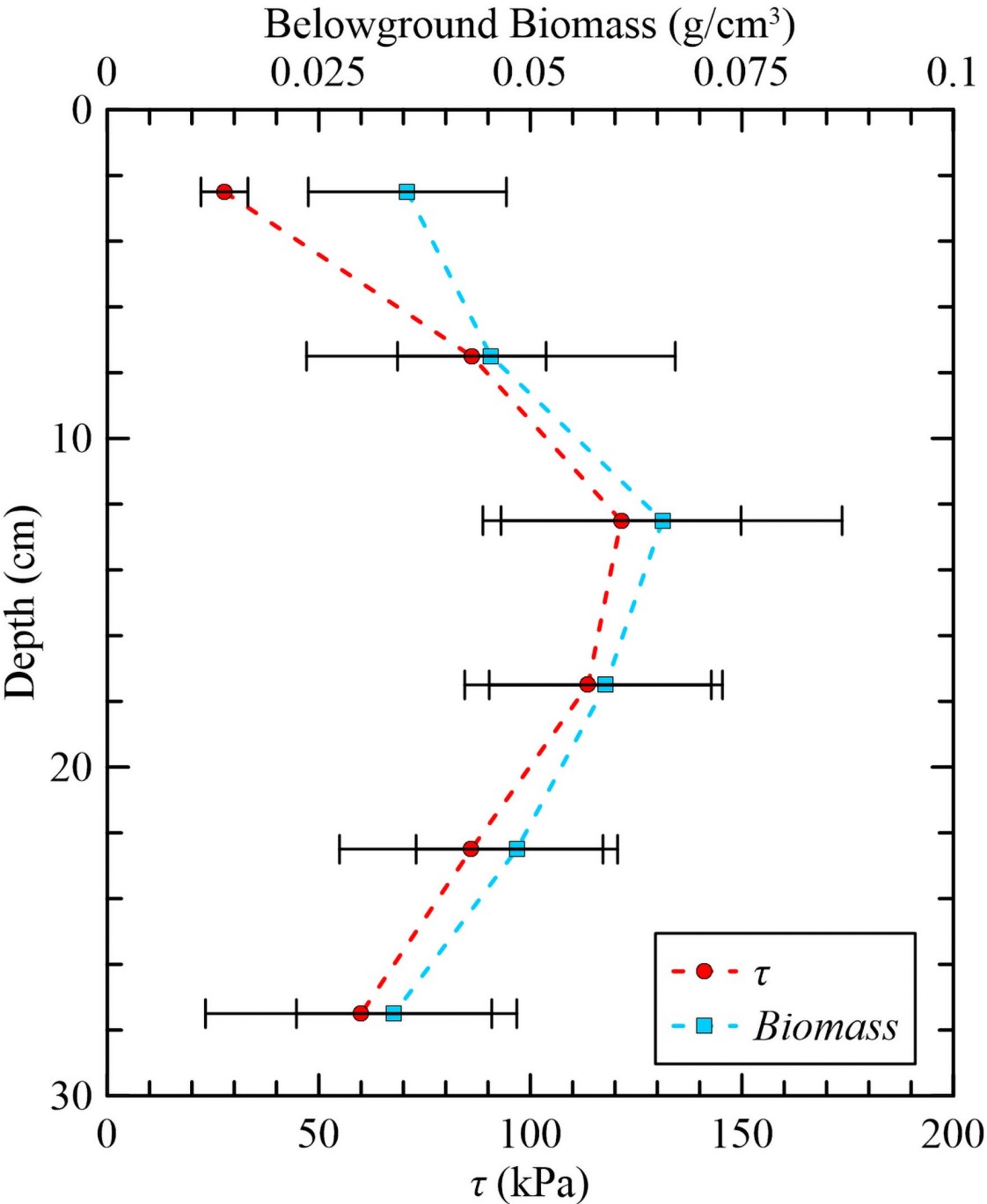

**Fig 9. Analogous trends between belowground biomass (g/cm³) and shear strength (kPa) with depth.** Horizontal bars are ±1 SD.

information of restorations. A key observation of Fig 11 is that the peak strength and below-ground biomass decrease with increasing TLP thickness, which suggests live roots and rhizomes may be providing a sharp increase in strength. Another observation from Fig 11 showed that as the thickness of TLP increases, the scatter of CPT also increased. This signifies the shear strength in the dredged sediment is significantly more variable because of a wide

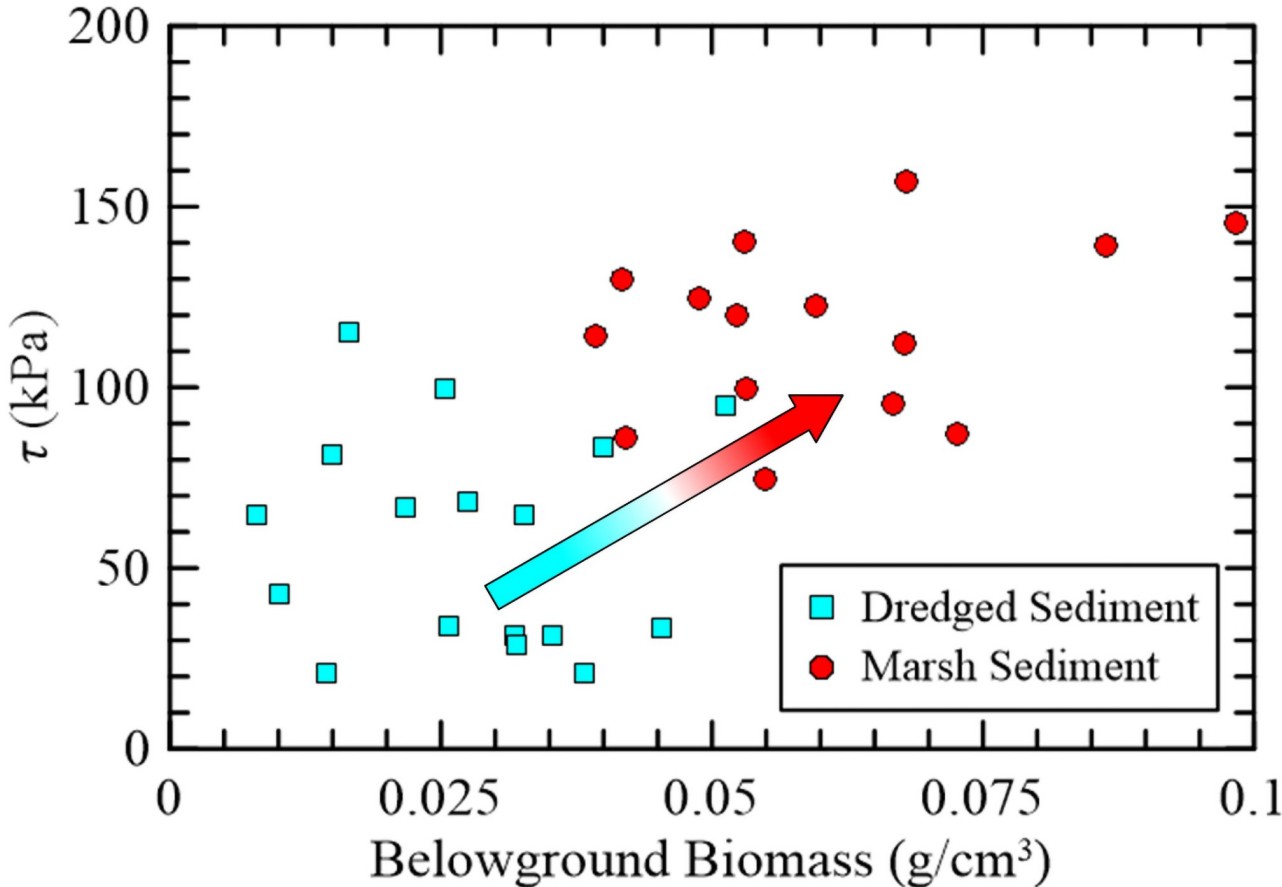

**Fig 10. Shear strength compared to belowground biomass measurements within the dredged sediment and marsh sediment samples.** Arrow illustrates the trajectory of dredged sediment to established marsh sediment.

range of particle gradation (sand to clay) and different levels of biomass productivity [23]. In addition, a transition in the belowground biomass profiles occurs at or near 10 cm of TLP, which could signify a tipping point that the geotechnical properties will resemble vegetated wetlands or inorganic sediments from the TLP [14]. This information can help guide future sediment nourishments by providing design constraints on upper limits of placement thicknesses and a methodology to quickly gauge the establishment of vegetation.

During the 2019 field investigation, tracks of ponded marsh were found throughout the area (Fig 1). CPTs were conducted within the ponded tracks (4) and the directly adjacent vegetated zones (6). The tip resistances ($q_t$) with depth for both conditions alongside a picture of the test area after a CPT was completed is shown in Fig 12. In the vegetated zones, the $q_t$ increases to a peak resistance before decreasing to the underlying soil. However, the ponded zones indicate that the peak $q_t$ occurs at the soil surface and then decreases with depth, exemplifying the site was disturbed through compaction. During the nourishment of the site, this was a common path for marsh buggies carrying the coir logs and dredging pipe and the effects of the continued compaction are still prevalent within the marsh 3.5 years later. It has been previously documented how compaction induced via heavy machinery during the restoration of a wetland results in negative impacts on root development and biomass production, which was attributed to the increase in bulk density inhibiting root penetration [37, 38]. This

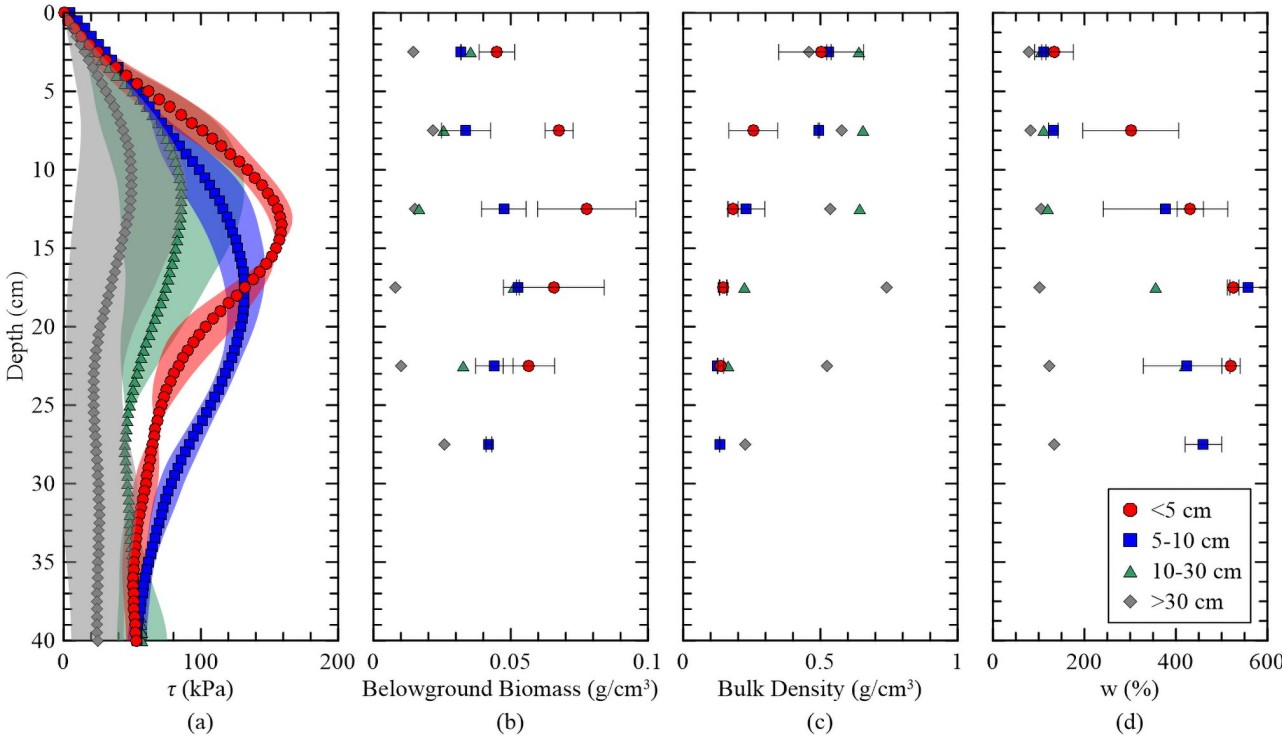

**Fig 11. The influence of dredged sediment depths across varying intervals.** (a) Shear strength, (b) belowground biomass, (c) dry bulk density, and (d) moisture content (%) at ≤5 cm (red), 5–10 cm (blue), 10–30 cm (green), and >30 cm (gray). Shaded regions and horizontal bars represent ±1 SD.

demonstrates how sensitive wetland environments are to anthropogenic disturbances and careful consideration is required during planning and construction to minimize lasting negative impacts like this.

The results of this study showed that within the nourished site, the shear strength values were higher than the tall-form of *S. alterniflora* at the reference site but were significantly less than the peak shear strength experienced within the short-form *S. alterniflora*. Additionally, the peak shear strength was found at a deeper depth than the two reference sites, which could signify that the vegetation roots have yet to fully establish themselves within the dredged sediment. Overall, the average underlying soil shear strength was 39 kPa for the nourished site. If the two mudflat sites were removed, the average increases to 45 kPa, which is consistent with the reference site. This shows that the underlying soil exhibits an intrinsic strength that is consistent throughout the nourished and control areas and not affected by surface wetland functions.

Prior to the TLP nourishment, this site was highly degraded as vegetated portions of wetlands rapidly transitioned to un-vegetated shallow pannes. This downward shift of elevation relative to sea-level induces higher levels of stress on the vegetation health, lowering the wetland resistance to physical stressors [25]. This is evident in the lower shear strength of the tall-form of *S. alterniflora* compared to the short-form, which resides at a higher elevation in the tidal regime (i.e. lower inundation periods). The rate of sea-level rise in this area is 1 cm/yr [39] while accretion is 0.3 cm/yr [40] leaving an elevation deficit of -0.7 cm/yr. The average elevation gain from this TLP nourishment increased the marsh elevation by 24 cm, indicating that this elevation deficit is offset by 34 years.

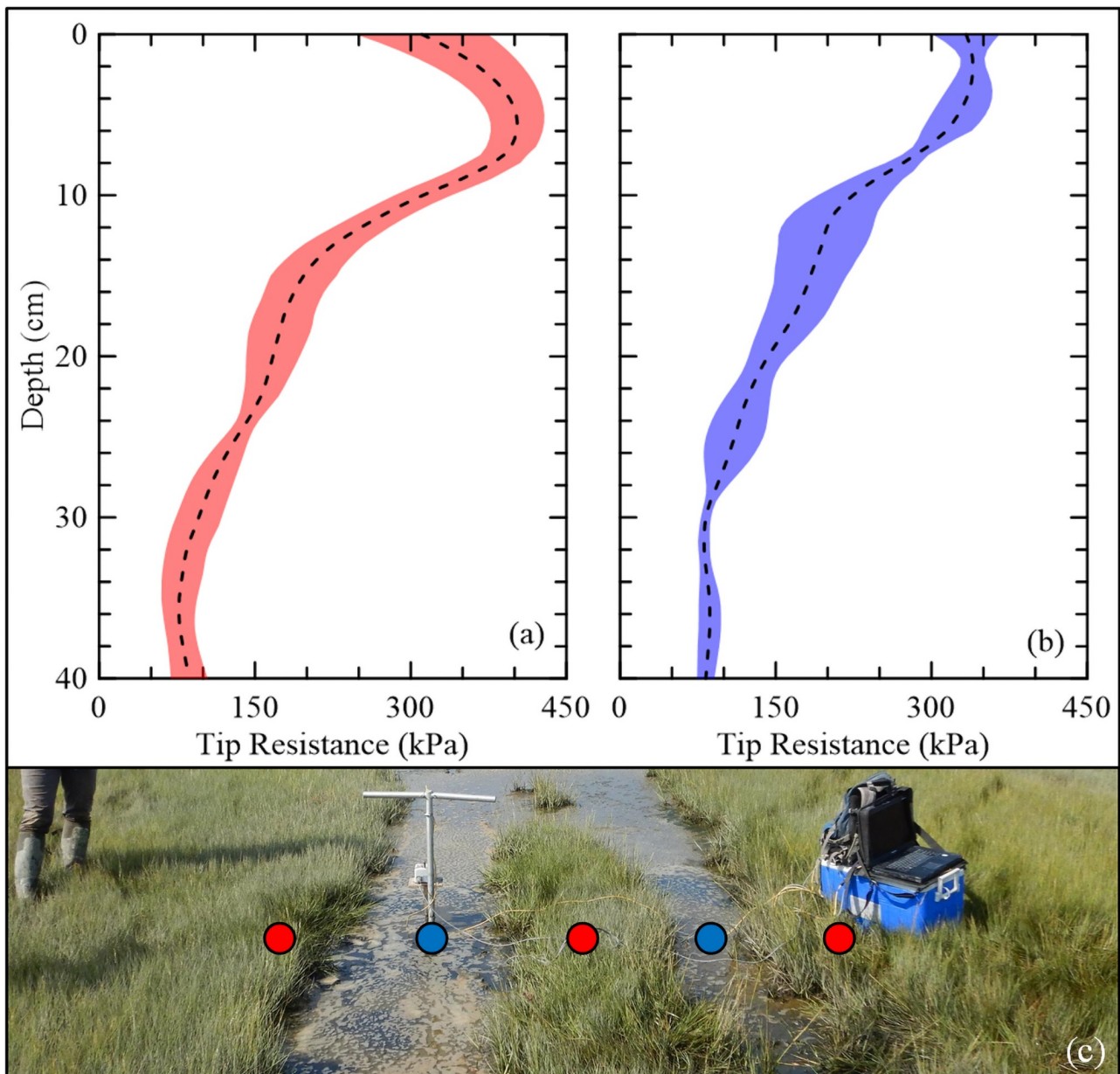

**Fig 12. Average tip resistances for the (a) vegetated and (b) ponded zones, and (c) testing locations across construction tracks.** Red and blue circles denote the vegetated and ponded zones, respectively. Shaded regions represent ±1 SD.

## Conclusions

This field investigation was performed using co-located cone penetrometer tests and soil cores to measure wetland strength establishment post-TLP nourishment. These measurements were compared to traditional performance measures (e.g., belowground biomass, bulk density, and moisture content) across two transects within a nourished wetland. The primary findings of this study are summarized as follows:

- The nourished site exhibited weaker shear strength than the reference site but vegetation establishment does appear to be occurring in the previously ponded areas.

- Belowground biomass and CPT shear strength measurements correlated with depth, demonstrating that this methodology can provide accurate quantifications on site development trajectory in a more efficient and less intrusive manner than traditional ecological and geotechnical techniques.

- The underlying soil below the roots exhibits an intrinsic strength that is consistent throughout both restored and reference areas and are not affected by surface wetland functions and is likely a function of the long-term geological history of the site.

- Sediment sampling directly post-nourishment and 3.5 years later showed a redistribution of grain size gradients, which can be attributed to coastal processes reworking the sediments, accretion, or bioturbation.

- Heavily used construction tracks compacted a small area of the marsh platform, inhibiting the establishment of vegetation 3.5 years post-construction.

For this specific case, baseline data would have provided additional information to better indicate wetland establishment trajectories and thus is highly recommended for future field investigations. The results presented herein capture one point in the establishment process of a sediment-nourished wetland, and future monitoring is necessary to fully understand the long-term impacts of TLP on marsh resilience and maintenance. While other meaningful studies utilize extended growing seasons to understand the long-term effects of sediment placement on the biological, chemical, and physical dependencies, this field investigation demonstrated the usefulness of the cone penetrometer when evaluating wetland strength establishment. The utilization of the cone penetrometer in combination with traditional performance measures can provide rapid assessments of wetland status for coastal restoration and management practices.

## Supporting information

**S1 Data.**
(XLSX)

## Acknowledgments

The authors would like to thank support offered by the Engineering Research and Development Center (ERDC) through the USACE Dredged Operations and Environmental Research (DOER) program, the Department of Defense (DoD) through its Science, Mathematics, and Research for Transformation (SMART), and the Resources and Ecosystems Sustainability, Tourist Opportunities, and Revived Economies of the Gulf Coast States Act (RESTORE Act) center of excellence grant. In addition, the authors would like to thank Jacob Berkowitz and Christine VanZomeren from ERDC's Environmental Lab and the following LSU students for assisting with field and laboratory testing: Dominion Ajayi, Mahajebin Haque, Cameron Markowitz, and Amina Meselhe. The individuals displayed in this manuscript have given written informed consent to publish these case details.

## Author Contributions

**Conceptualization:** Brian D. Harris, Donnie J. Day.

**Data curation:** Brian D. Harris.

**Formal analysis:** Brian D. Harris, Donnie J. Day, Jack A. Cadigan.

**Investigation:** Brian D. Harris, Donnie J. Day, Susan E. Bailey, Zachary J. Tyler.

**Methodology:** Brian D. Harris, Donnie J. Day, Jack A. Cadigan.

**Supervision:** Navid H. Jafari.

**Visualization:** Brian D. Harris.

**Writing – original draft:** Brian D. Harris, Donnie J. Day, Jack A. Cadigan, Navid H. Jafari, Susan E. Bailey, Zachary J. Tyler.

**Writing – review & editing:** Brian D. Harris, Donnie J. Day, Jack A. Cadigan, Navid H. Jafari, Susan E. Bailey, Zachary J. Tyler.

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
