## [Decision Letter · Decision Letter 0]

16 Oct 2020

PONE-D-20-25909

Establishment of Soil Strength in a Nourished Wetland using Thin Layer Placement of Dredged Sediment

PLOS ONE

Dear Dr. Harris,

Thank you for submitting your manuscript to PLOS ONE. After careful consideration, we feel that it has merit but does not fully meet PLOS ONE’s publication criteria as it currently stands. Therefore, we invite you to submit a revised version of the manuscript that addresses the points raised during the review process.

In the process of making your revisions, please pay careful attention to the comments from Reviewers 2 and 3 about identifying the goals/objectives of the project, broadening the scope by referencing additional literature from a broader geographic range, and using standard (or less-regionally specific) terminology that is common to the field of wetland restoration. By addressing their comments, there is an opportunity to reframe, and more clearly link, the introduction and discussion sections and to clarify restoration goals and metrics for success.

We look forward to receiving your revised manuscript.

Kind regards,

Julia A. Cherry

Academic Editor

PLOS ONE

Journal Requirements:

2.  Thank you for including the following ethics statement on the submission details page:

'Field work was conducted on land overseen by the Wetlands Institute (Avalon, NJ).

Contact: Lenore Tedesco'

* In the Methods section of your manuscript, please amend the ethics statement to confirm that the name institution approved the field site access and sampling collection.

Support for this research was provided in part by the Engineering Research and Development

451 Center (ERDC) through the USACE Dredged Operations and Environmental Research (DOER)

452 program, the Department of Defense (DoD) through its Science, Mathematics, and Research for

453 Transformation (SMART), and the Resources and Ecosystems Sustainability, Tourist

454 Opportunities, and Revived Economies of the Gulf Coast States Act (RESTORE Act) center of

455 excellence grant. In addition, the authors would like to thank Jacob Berkowitz and Christine

456 VanZomeren from ERDC’s Environmental Lab and the following LSU students for assisting with

20

field and laboratory testing: Dominion Ajayi, 457 Mahajebin Haque, Cameron Markowitz, and Amina

458 Meselhe.".

i) We note that you have provided funding information that is not currently declared in your Funding Statement. However, funding information should not appear in the Acknowledgments section or other areas of your manuscript. We will only publish funding information present in the Funding Statement section of the online submission form.

ii) Please remove any funding-related text from the manuscript and let us know how you would like to update your Funding Statement. Currently, your Funding Statement reads as follows:

 "The author(s) received no specific funding for this work.".

4. We note that Figures [1 and 5] in your submission contain [map/satellite] images which may be copyrighted. All PLOS content is published under the Creative Commons Attribution License (CC BY 4.0), which means that the manuscript, images, and Supporting Information files will be freely available online, and any third party is permitted to access, download, copy, distribute, and use these materials in any way, even commercially, with proper attribution. For these reasons, we cannot publish previously copyrighted maps or satellite images created using proprietary data, such as Google software (Google Maps, Street View, and Earth). For more information, see our copyright guidelines: http://journals.plos.org/plosone/s/licenses-and-copyright.

1.     You may seek permission from the original copyright holder of Figure(s) [1 and 5] to publish the content specifically under the CC BY 4.0 license.  

5. Please amend either the title on the online submission form (via Edit Submission) or the title in the manuscript so that they are identical.

6. We note that Figure [2] includes an image of a  participant in the study. 

8. Please include a separate caption for each figure in your manuscript.

Additional Editor Comments (if provided):

This manuscript on the establishment of soil strength using thin layer placement requires major revisions to address overarching concerns and specific comments identified during the review process. Please carefully consider the comments and recommendations for revision that the reviewers have made, especially as they pertain to the scope and terminology used to describe the project and its implications for wetland restoration. For instance, utilizing standard terminology for nourishment and thin layer placement projects and expanding the literature cited to encompass broader geographic range will help reach the broader audience of Plos One. Such revisions may necessitate a change in title as well. There are also a series of questions and comments about the site description and methods that need to be addressed to improve quality. And finally, it is crucial that the objectives of the project (TLP evaluation relative to reference and use of CPT to measure success) be clarified throughout the paper and substantiated by the data. Reviewer 3 makes some useful suggestions in this regard. Addressing suggestions from these reviews will greatly improve the clarity of the paper and its potential impact in the field of wetland restoration, while also expanding the scope to be more suitable for the international readership of the journal.

Reviewers' comments:

Reviewer's Responses to Questions

**Comments to the Author**

1. Is the manuscript technically sound, and do the data support the conclusions?

Reviewer #1: Partly

Reviewer #2: Partly

Reviewer #3: Yes

2. Has the statistical analysis been performed appropriately and rigorously? 

Reviewer #1: Yes

Reviewer #2: N/A

Reviewer #3: N/A

3. Have the authors made all data underlying the findings in their manuscript fully available?

Reviewer #1: Yes

Reviewer #2: Yes

Reviewer #3: Yes

4. Is the manuscript presented in an intelligible fashion and written in standard English?

Reviewer #1: No

Reviewer #2: Yes

Reviewer #3: Yes

5. Review Comments to the Author

Reviewer #1: Improvement of English is suggested. For example, the article mentions "At the time of investigation" on several occasions. Rewriting of those sentences is suggested. Also, the article has issues associated with same word once UPPER CASE and once LOWER CASE. Some of the references were not written appropriately. Mixing of present and past tense. The pdf should contain the suggested corrections. Please check those.

Reviewer #2: Overall, I think this is an interesting article that should be accepted for publication in PLOS One pending major revision as detailed in the comments below. The manuscript addresses the impacts on sediment parameters of a thin layer placement. Sea level rise is a challenge faced by wetlands globally, thus evaluating strategies to help marshes keep pace with sea level rise is important. Despite the importance of the topic, this paper needs major revision before being acceptable for publication in PLOS One. The major and minor issues that I found are documented below.

My major comments include

1) Introduction – I think that the overall introduction of this paper needs to be broadened in scope, terminology, and literature review to be more applicable to the global audience of PLOS One.

a. PLOS One is read by a wide range of scientists, not just engineers so I think that some of the background need broader focus, and I have tried to suggest places below (in specific comments) where I feel those additional references or information is needed.

b. A lot of the terminology used (esp. nourishment) is regionally specific.

c. Most of the literature discussed is older and focused on just a couple of ecosystems in the United States (Gulf of Mexico and East Coast). This should be broadened to include more recent studies (e.g. Cahoon et al. 2019 Estuaries and Coasts) and more diverse locations (e.g. Thorne et al. 2019 Ecol Eng from CA, Wigand et al. 2017 Estuaries and Coasts).

2) Methods – Overall, there is a lot of repetitiveness and misplaced information between the Methods and Results. Both sections should be cleaned up and better organized. In addition, the site and experiment description need to clearer as I lost track of what a region versus a site versus a transect versus a TLP area was.

a. The figures and tables are emphasized in the sentence structure instead of the result being discussed with a parenthetic reference to the table or figure that presents the data.

b. A lot of the text in the Results seems to just describe the numbers from the figures which I found confusing. I could see the numbers on the figures and would prefer to have the pattern described in the Results text.

c. Several key methods about vegetation cover and grain size are missing. s

3) Figures – I found several of the figures not as clear as could be due to missing labelling or need of additional details. These are discussed in detail below.

4) Throughout the manuscript, I felt as if the parameters (shear strength) measured were done well but then how broadly these could be applied were overstated. CPT strength measurements may correlate with the measured parameters but it is overstated to claim that they predict restoration success or the entire ecological restoration trajectory. I think being more clear in what is measured at the start and what is inferred would improve the manuscript.

Abstract

L. 59 – This is really not establishment of soil strength but redevelopment of (post-TLP) or changes in.

L. 60 – What are establishment markers? This is too vague for an important point in the abstract.

L. 60 – I think this should read “lower strengths”, not “weaker strengths” which is confusing.

L. 64 – 66 (Starting During the construction process) – This sentences seems unnecessary and like an anecdote, not an abstract sentence.

L. 61 – The root system? Whose roots? Vegetation?

L. 64 – Trajectories is also too vague. You should state vegetation or belowground biomass specifically because there are a lot of restoration trajectory parameters (e.g. invertebrates, algae) that you are not measuring.

Introduction

L. 74 – Habitats should not be plural.

L. 76 hurricanes have accelerated

L. 77 – I think is more true if written “loss, leading to need for comprehensive coastal restoration management plans” so this did not always happen.

L. 81 – 83 – This statement is not always true – it is hypothesized that TLP can be conducted without ecological implications but that varies with location and impacts measured and thickness of sediment applied. A more in-depth literature review is required here. This is also a sentence that needs more modern and more geographically varied references (as discussed above).

L. 83-85 – There are more than just this method to conduct TLP – if you are going to discuss this you should acknowledge other methods or drop this sentence.

L. 91-93 – This repetitive of earlier paragraph.

L. 98 – Spartina-dominated

L. 99 – Where were these experiments? Location and type of marsh would be useful for understanding the context of this reference.

L. 101-102 – You also need to expand this discussion or delete this sentence.

L. 103 – 105 _ I would split this into two sentences after the reference as right now it has a misplaced modifier.

L. 106 – Delete establishment as I think this is confusing.

L. 108 – What do you mean by physical index properties?

L. 109 – You refer to the dredge outfall, but we need more information. Is this permanent or just for construction?

L. 111 – 113 – We also need more information about the reference site to evaluate its choice as a comparison. It is unfortunate that you are missing sediment cores from there but other site parameters will help justify its choice as a reference site.

L. 113 – This is an example of where I feel the implications of this work are overstated. This will not define wetland restoration success but will instead advance our understanding of how CTP relates to specific biomass parameters. You can make the argument (which you should) that AG and BG biomass are related to important ecosystem development metrics.

Background

Figure 1 – This figure needs larger context. I would like to see the inset be expanded to show more of the East Coast, not just a section of NJ and DE. This is hard for a non-local or global reader to understand.

L. 120 – On the bay side of what? And from Figure 1, it appears as if is actually on the Oceanside? Please clarify

L 120 – the wetland – which wetland? The TLP site or the reference site or the whole thing?

L. 127 – When was the area identified as transitioning? The idea of this rapid transition is mentioned several times throughout the paper but not enough detail on the time frame and the observed transition is presented to completely understand.

Figure 1b is useful.

L. 131 onwards in this paragraph – The methods behind these measurements needs to be explained. In fact, these data are repeated in different places. I think the methods for this survey need to be explained in the Methods, and these data need to be presented in the Results.

L. 134 – This is an example of the terminology confusion – what is areas here? Is this the shallow pannes discussed above (L. 132)?

L. 134 – Again my comment about rapid transition. Over what time period?

L. 144 – This seems repetitive of information above or I do not understand the terms for your site versus marsh etc.

Methods

L. 150 – Another landscape term is introduced here “cells”. These should appear on the map.

L. 151 – I would rename “Interior”

L. 153 – 154 – This sentence is confusing. Do both transects begin at the dredge outfall or do they begin at different starting points? Or do you mean that the two transects have different elevations and starting habitats?

L. 157 – Write out the reference name in this case.

L. 159 – I think the more accurate term is abiotic, not ecological.

L. 166 – What four tests? Are these pilot evaluations?

L. 174 onwards should be moved to Results.

L. 188 – Again what are sites? The TLP versus reference marshes? Or the points along the transect?

L. 190 – Here is an example of leading with the Figure instead of stating the results and referring to the figure.

L. 195 – Was the vegetation rinsed to remove mud?

Results

L. 200 – 207 – This should be moved to Methods

L. 209 – The amount of elevation loss over what time period?

L. 201 – Here another landscape term is introduced – region. While these are shown on the map, it is confusing to understand how cell, site, region all fit together in the sampling scheme.

L. 219 – 222 – These sentences have a lot of interpretation and belong in the Discussion.

L. 223 – 227 – These sentences belong in methods.

L. 225 – No grain size methods were discussed at all in the Methods. This needs to be added.

L. 227 – Post-nourishment

L. 227 – I am not sure what is meant by an increase? I think it just needs to be reworded to be clearer.

L. 231 = This belongs in the Discussion, and this is an example of what I find overstated. We don’t know that this is vegetation-driven accretion from your study.

Headings of Establishment here confused me – establishment of what?

L. 234 – 236 – This is already described in the Methods so I think could be deleted.

L. 235 – Vegetation methods need to be described better in Methods (as discussed above) so this should not be “appeared to be” but an actual community analysis or description.

This paragraph has lots of examples of leading with the Figure or Table when the results should be summarized.

L. 240 – Shear strength increased down the core or over time?

This whole section is difficult for me to read – I think the figures speak for themselves and I would prefer to see the pattern described without the exact numbers. With a well-designed figure, I can read the exact numbers myself.

L. 240 – At the time of investigation? Tell us when it was.

Table 1 is not vegetation but AG and BG biomass

L. 268 – Again the opening sentences are repetitive of information from earlier and should be deleted.

L. 296 – More attention needs to be paid to the missing data from Reference site. I understand that things happen but it could affect conclusions.

L. 296 – Dominant, not dominate

Compaction – This section does not contribute much to the manuscript in my mind so either needs to be better linked and connected to the story or deleted.

L. 373 – This is missing something….

Discussion

L. 381 – This is still one of the points on which I am confused. When are we talking about specifically? Why was the site so degraded? Salt pannes can be natural so I am not convinced that the transition to salt pannes alone was bad or a sign of degradation.

While I like linking this back to ecological theory, I am not convinced of the stage model as it relates to your data and conclusions.

Reviewer #3: This manuscript addresses soil strength and biophysical factors of thin layer placement as a restoration technique for degraded marshes. The manuscript describes the use of a cone penetrometer as a proxy for soil strength along two transects, a heavy machinery impacted zone, and a control area. Overall, the manuscript addresses important data gaps in methodology and marsh restoration trajectories following thin layer placement. However, the manuscript requires additional clarification and text in several locations, and mainly in the discussion section. I am recommending major revisions to this manuscript, which upon completion will provide important new data to the wetland literature. Specific comments are provided below.

60 Clarify “traditional wetland establishment markers” to specify belowground biomass, bulk density, and moisture content. Are these parameters only related to wetland establishment, or more broadly applicable to wetland condition?

61 This statement is a little misleading. Based on the site description, large areas were un-vegetated prior to thin layer placement. Post thin layer placement, these areas are beginning to vegetate. Is a comparison between a vegetated control area and an un-vegetated open treatment area valid?

80 The use of “nourishment” is also misleading. Are you nourishing elevation? The objective of the thin layer placement is to increase elevation. A secondary effect is nutrient addition (i.e. “nourishment”) from the dredged material.

85 There are examples of thin layer placement on the West coast, and should be included in this section. Specifically, Seal Beach, CA is an example of a West Coast TLP project.

92 Is there really a significant contribution of organic material from the dredged sediment?

102 Add a transition here; the change in ecological discussion to shear strength is abrupt.

129 The literature uses “short form” and “tall form” of S. alterniflora.

154 Establishment of what?

157 Is ultra-soft wetland soils analogous to unconsolidated soils? If so, include unconsolidated soils as an additional term that people are familiar with. If not, define what “ultra-soft” refers to.

204 Consolidation of the dredged material is expected. However this is not addressed until much father in the text. Recommend this information is moved up in the text to clarify that elevation loss was not un-expected.

231 Vegetation driven accretion? The accumulation of organic material through vegetation growth redistributes fine materials? More likely redistribution of fine material is through hydrodynamic processes and bioturbation; above ground biomass would increase fine material deposition. Clarification of this statement is needed.

265 Table 1. What does Vegetation Depth (cm) describe? The original marsh surface? Clarification is needed.

326 Clarify what “dredge sediment is further established” means. Vegetation establishment within the placement sediment?

361 Correct biomass to organic material.

365 Tracks of what?

308 The discussion section is very thin. One important point to discuss is open water areas that received dredged material in the context of the vegetated control areas. These are not directly comparable, yet offer an opportunity to discuss the trajectory of an open water area that receives dredge material moving towards shear strength of a vegetated area. Additionally, a discussion of Avalon marsh trajectory is missing, and perhaps is not possible given one sampling date? How does this fit into the conceptual model? Going back to the objectives outlined in 1121-115, was this wetland restoration project a “success”? How do you define success with one sampling date with CPT? There seems to be two competing objectives of this paper: 1. how does the TLP site compare to the reference area and 2. use of the CPT as a methodology to measure restoration success and trajectory. Revise the discussion section, and throughout as needed, to address these two objectives.

387 Does this calculation consider additional potential elevation gains from organic matter accumulation and resulting accretion from healthy vegetation growth?

6. PLOS authors have the option to publish the peer review history of their article (what does this mean?). If published, this will include your full peer review and any attached files.

Reviewer #1: No

Reviewer #2: No

Reviewer #3: No

---

## [Author Response · Author response to Decision Letter 0]

8 Mar 2021

A complete response to each comment has been provided in the accompanying word document. Thank you!

---

## [Decision Letter · Decision Letter 1]

27 Apr 2021

Establishment of Soil Strength in a Nourished Wetland using Thin Layer Placement of Dredged Sediment

PONE-D-20-25909R1

Dear Dr. Harris,

We’re pleased to inform you that your manuscript has been judged scientifically suitable for publication and will be formally accepted for publication once it meets all outstanding technical requirements.

Kind regards,

Julia A. Cherry

Academic Editor

PLOS ONE

Additional Editor Comments (optional):

Thank you for your careful attention to the reviewers' comments.

Reviewers' comments:

Reviewer's Responses to Questions

**Comments to the Author**

1. If the authors have adequately addressed your comments raised in a previous round of review and you feel that this manuscript is now acceptable for publication, you may indicate that here to bypass the “Comments to the Author” section, enter your conflict of interest statement in the “Confidential to Editor” section, and submit your "Accept" recommendation.

Reviewer #2: All comments have been addressed

2. Is the manuscript technically sound, and do the data support the conclusions?

Reviewer #2: Yes

3. Has the statistical analysis been performed appropriately and rigorously? 

Reviewer #2: Yes

4. Have the authors made all data underlying the findings in their manuscript fully available?

Reviewer #2: Yes

5. Is the manuscript presented in an intelligible fashion and written in standard English?

Reviewer #2: Yes

6. Review Comments to the Author

Reviewer #2: I appreciate the careful revisions of this manuscript as well as the detailed response to reviewers. The revisions dramatically improve the manuscript and make it both more understandable as well as more applicable to a variety of readers.

7. PLOS authors have the option to publish the peer review history of their article (what does this mean?). If published, this will include your full peer review and any attached files.

Reviewer #2: No

---

## [Editor Report · Acceptance letter]

29 Apr 2021

PONE-D-20-25909R1 

Establishment of Soil Strength in a Nourished Wetland using Thin Layer Placement of Dredged Sediment 

Dear Dr. Harris:

I'm pleased to inform you that your manuscript has been deemed suitable for publication in PLOS ONE. Congratulations! Your manuscript is now with our production department. 

Kind regards, 

on behalf of

Dr. Julia A. Cherry 

Academic Editor

PLOS ONE